# Night shift work, chemical coexposures and risk of female breast cancer in the Norwegian Offshore Petroleum Workers (NOPW) cohort: a prospectively recruited case-cohort study

Fei Chih Liu ,[1,2] Marit Bragelien Veierød,[2] Kristina Kjærheim,[1] Trude Eid Robsahm,[1] Reza Ghiasvand,[1,3] H Dean Hosgood,[4] Sven Ove Samuelsen,[5] Magne Bråtveit,[6] Jorunn Kirkeleit,[6] Nathaniel Rothman,[7] Qing Lan,[7] Debra T Silverman,[7] Melissa C Friesen,[7] Ronnie Babigumira,[1] Nita Shala,[1,2] Tom K Grimsrud,[1] Jo Steinson Stenehjem [1]

For numbered affiliations see end of article.

**Correspondence to**
Dr Jo Steinson Stenehjem;
jo.stenehjem@kreftregisteret.no

## ABSTRACT

**Objectives** This study examined the association between night shift work and risk of breast cancer, overall and by hormone receptor subtype, among females in the Norwegian Offshore Petroleum Workers (NOPW) cohort. We also examined the association of coexposure (chlorinated degreasers and benzene) and breast cancer risk, and possible interaction with work schedule.

**Design** Prospectively recruited case-cohort study within the NOPW cohort.

**Setting** Female offshore petroleum workers active on the Norwegian continental shelf.

**Participants** 600 female workers (86 cases and 514 non-cases) were included in the study. We excluded workers that died or emigrated before start of follow-up, had missing work history, were diagnosed with breast cancer or other prior malignancy (except non-melanoma skin cancer) before start of follow-up.

**Results** No overall association was found between breast cancer risk and work schedule (HR 0.87, 95% CI 0.52 to 1.46 for work schedule involving night shift vs day shift only). There was no significant association between work schedule and risk of any breast cancer subtype. No significant interactions were found between work schedule and chemical coexposures (breast cancer overall $P_{\text{interaction chlorinated degreasers}}$=0.725 and $P_{\text{interaction benzene}}$=0.175).

**Conclusions** Our results did not provide supporting evidence that work schedule involving night shift affects breast cancer risk in female offshore petroleum workers, but should be considered cautiously due to few cases. Further studies with larger sample sizes are warranted.

## INTRODUCTION

Breast cancer is the most frequent cancer among females and the incidence is higher in industrialised compared with low-income and middle-income countries.[1] The increase seen in industrialised societies has mainly been attributed to reproduction and lifestyle

### Strengths and limitations of this study

► Complete cancer and population registry data ensure complete follow-up.
► Comprehensive work history information of night/rollover shift duration for up to eight employments per worker.
► Job-exposure matrices of chemical exposure developed specifically for cancer studies in the cohort.
► Females in the cohort were young at baseline, which yielded relatively few breast cancer cases.
► We lacked information on family history of breast cancer, exogenous hormone use and chronotype.

factors.[2 3] However, the use of artificial light during work at night, leading to disruption of the internal circadian clock and suppressed melatonin production, may accelerate the transformation of normal cells to malignant mammary cells and promote the risk of breast cancer.[4 5]

Norwegian offshore workers employed at oil and gas installations have an extreme touring work pattern.[6] A standard 2-week tour on Norwegian installations is minimum 12 hours of work 7 days per week. The work schedule is either 14 day shifts, 14 night shifts or 14 rollover shifts (starting with 7 days followed by 7 nights or vice versa).[6] During a night shift, offshore workers are constantly exposed to artificial light.[6] These extreme working hours meet the definition of night shift work (≥7 hours of work including the period from midnight to 05:00 hours).[7 8]

The relationship between night shift work and breast cancer is still unclear. Several studies have reported a positive association

between night shift work and breast cancer,[9–12] while others did not.[13–16] A challenge has been the lack of high-quality exposure data on light during night time and detailed data on night shift work schedules.[4] A recent pooled analysis of five population-based case–control studies with complete work history reported an increased risk of breast cancer in premenopausal females who had high intensity and long duration of night shift work.[17] Further, night shift work may be associated with an increased risk of oestrogen receptor (ER)-positive, progesterone receptor (PR)-positive or human epidermal growth factor receptor 2 (HER2)-positive breast cancer.[18–21] However, night shift work has also been reported to increase the risk of ER-negative breast cancer,[22] and whether the association between night shift work and breast cancer varies by subtype remains unclear.

Based on the limited evidence for the carcinogenicity of night shift work, the International Agency for Research on Cancer (IARC) recently reconfirmed night shift work as a probable carcinogen to humans (group 2A),[23] asking for improved exposure assessment in future studies to reduce heterogeneity between studies and allow for better comparisons. IARC also encouraged better reporting of potential co-exposures to occupational carcinogens. On offshore petroleum installations, some female workers may be exposed to a number of carcinogens, but in particular benzene during drilling, production and maintenance,[24 25] and historically chlorinated degreasers during cleaning of metal parts (trichloroethylene) or dry-cleaning of work clothes (tetrachloroethylene).[21] Cohort and population-based studies have linked benzene and tetrachloroethylene to increased breast cancer risk,[26 27] and recently, night shift work has received increased attention as a possible effect modifier of the association between chemical exposure and cancer by desynchronising detoxification mechanisms.[28 29] To our knowledge, chemical exposure and night shift work have not been examined simultaneously in relation to breast cancer risk.

The Norwegian Offshore Petroleum Workers (NOPW) cohort holds information on work schedule for each individual employed 1965–1998 and job exposure matrices (JEMs) capturing chemical exposures in the offshore work environment.[30] The aim of this study was to examine the association between shift work and breast cancer risk, overall and by receptor status subtypes, among prospectively recruited females in the NOPW cohort. We also examined the association of chemical exposures (chlorinated degreasers and benzene) and breast cancer risk, and possible multiplicative interactions with work schedule.

## MATERIALS AND METHODS
### Patient and public involvement
This study was conducted without patient involvement.

### Study population and study design
The NOPW cohort was established in 1998 by the Cancer Registry of Norway (CRN). Questionnaires were filled in by current or former male and female offshore petroleum workers with a minimum of 20 days work on the Norwegian continental shelf between 1965 and 1998.[30] In total, 27 917 workers were included (estimated response rate 69%) in the NOPW cohort.[31] Details on the cohort and its establishment have been published elsewhere.[30 31] All workers filled in an informed consent for participation in the study.

We used a stratified case-cohort design[32] as complete information on work history had to be extracted manually from the questionnaires for breast cancer cases and a random subsample of the cohort (hereafter the 'subcohort'). The case-cohort design allows prospective models such as Cox-regression, and the subcohort may be used as controls for different types of cancer cases, as it is not matched to specific cases.

### Identification of cancer cases
The NOPW cohort was linked to the CRN and the National Population Register for information on cancer, death and emigration, by using the unique personal identification numbers assigned to all Norwegian citizens.[33] Follow-up started 1 July 1999 and ended 31 December 2017. Reporting of incident cancers to the CRN is mandatory in Norway and the degree of completeness and validity is high, with morphology verified for 99.3% of the breast cancers.[33] Cancer cases were defined according to the International Classification of Diseases, 10th Revision, C50 for breast cancer. Immunohistochemistry and pathology reports were submitted routinely to the CRN to assess breast cancer receptor status.

### Study sample
A total of 25 347 males were excluded from the full cohort, and breast cancer cases and the subcohort were identified among the remaining 2570 female offshore workers (figure 1). Among the cases, we applied the exclusion criteria (1) death or emigration before start of follow-up (n=0) (2) missing work history (n=10), (3) breast cancer before start of follow-up (n=7) and (4) other prior malignancy except non-melanoma skin cancer before start of follow-up (mainly squamous cell carcinoma as the CRN does not routinely record information on basal cell carcinoma of the skin (n=0)), resulting in 86 breast cancer cases in the analyses (figure 1).

The initial subcohort (n=557) was sampled from the female cohort (n=2570) and was established in two steps to secure at least five non-cases per case. First, we oversampled (ie, selected all) the female cohort members born 1920–1939 and 1975–1979 (n=102) to secure that the oldest and the youngest female non-cases were comparable to the cases in these birth cohorts. Second, for those born 1940–1974, we drew at random within strata of 5-year birth cohorts 455 female workers, who were frequency matched to the birth year distribution of all potentially occupational cancers estimated for 2017 (n=91). We excluded 43 females that died or emigrated before start of follow-up (n=1), had missing work history

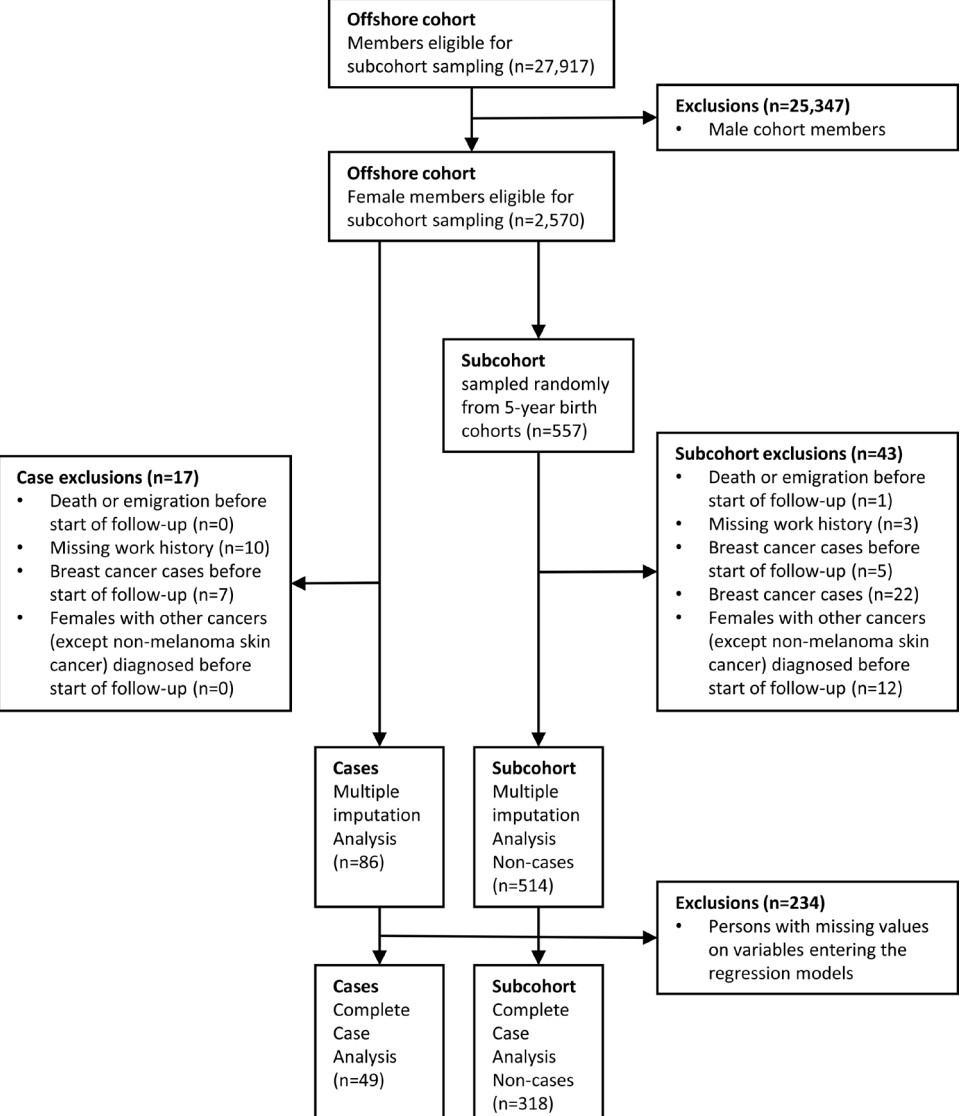

**Figure 1** Overview of study design and exclusions.

(n=3), were diagnosed with breast cancer (n=5) or other prior malignancy (except non-melanoma skin cancer (n=12)) before start of follow-up, or were diagnosed with breast cancer during follow-up (cases, n=22). Thus, for our analyses, 514 females remained in the subcohort (figure 1).

### Exposure assessment
#### Work schedule
We handled work history data of 514 female subcohort members and 86 cases who reported up to eight employments each. The process of harmonising overlapping employment records was handled by collapsing jobs within the same category and by splitting jobs of different categories into proportionally equal parts, according to a method described previously.[34]

The self-reported work schedule for each job was recorded as either day, night or rollover shift for 2-week tours. Day shift is equivalent to 14 consecutive days of daytime work usually between 07:00 and 19:00 hours. Night

shift is equivalent to 14 consecutive days of night-time work usually between 19:00 and 07:00 hours. Rollover shift is equivalent to seven consecutive days of day-time work (same working hours as for day shift) then alternate to seven consecutive days of night-time work (same working hours as for night shift), or vice versa.

Some worked only day, but none worked only night or only rollover. Therefore, the variable work schedule was categorised as (1) day work only, (2) mixed I (some night/rollover +mainly day), (3) mixed II (mainly night/rollover +some day) and (4) missing. We then collapsed categories 2 (mixed I) and 3 (mixed II) into exposed (night/rollover shift) to represent workers that had ever worked some degree of night-time. For each individual, the duration of night/rollover shift was calculated by (1) cumulating the total number of years with night shifts, (2) cumulating the total number of years with rollover shifts, (3) summarising the number of years worked with night and rollover shifts into a combined continuous

variable and (4) categorising this variable according to median (unexposed, <1–6, >6) or quartile (unexposed, <1–2, >2–6, >6–10, >10) of years of night/rollover shift. Total employment duration was categorised according to quartiles (0–1.9, 2–5.9, 6–10.9, 11–24), expressed in years.

### Chemical coexposures and main occupational activity in last position

A research group at the University of Bergen has developed expert-based JEMs for the NOPW cohort to identify and describe the degree of exposure to agents, mixtures or exposure situations with known or suspected carcinogenic potential among offshore workers on the Norwegian continental shelf 1970–2005.[25 35] The JEMs and their development have been described in detail elsewhere.[25 36] In brief, due to lack of exposure measurements, the assessment of the JEM for chlorinated degreasers was probability-oriented and based on summary documents (eg, from company visits/interviews, risk assessment reports, sampling reports, product data sheets) on the exposure to trichloroethylene and tetrachloroethylene. Eight experts individually categorised the likelihood (unlikely, possible, probable) of chlorinated degreaser exposure (ie, products containing trichloroethylene and tetrachloroethylene) for 27 job categories.[35] Importantly, the exposure assessment of chlorinated degreasers was evaluated as 'possible' for catering workers, meaning that some workers may have been exposed while the probability of exposure was low. For the benzene JEM, measurement data were used to construct a time-line of exposure and to rate exposure intensity of tasks specific to each job category. Then, task duration and frequency were combined to create a semi-quantitative benzene exposure burden score specific to job category and time period (1965–1969, 1970–1979, 1980–1989, 1990–1998).[36] For each of the two agents, we generated a dichotomised variable indicating never or ever exposure.

Main occupational activity in each worker's last position was recorded as production and process, drilling and well maintenance, maintenance/inspection/deck construction, catering/office/administration and miscellaneous.

### Covariates

Age at first child and number of children were recorded in the questionnaire, categorised as 0, 1, 2, ≥3 children. Females aged ≥53 years at baseline in 1998 were assumed postmenopausal, according to the convention from the Million Women Study[37] which also was adopted by the Norwegian Women and Cancer study[38] that is representative of the Norwegian female population. In supplemental analyses, we also used age ≥51 as cut-point to define postmenopause (online supplemental tables S1 and S2), which also has been used in previous study.[39] Education was recorded as compulsory, vocational, folk high school, upper secondary (the two latter collapsed into upper secondary) and university/college.

### Data analysis

Cox regression, adapted to a stratified case-cohort design,[32] was used to estimate HRs of breast cancer and 95% CIs associated with work schedule, chlorinated degreasers and benzene. Cases were assigned a weight of 1 and subcohort non-cases were given weights according to the inverse sampling fraction in their corresponding 5-year birth cohort stratum. For those born 1940–1974, we drew at random within the 5-year birth cohorts and calculated a sampling fraction, defined as random sample/total cohort member, for each birth cohort. We then assigned a weight, which was defined as 1/sampling fraction, for each birth cohort and those weights were specified as offset in the Cox regression models. In the analyses, age was used as the time-scale with subjects entering start of follow-up on 30 June 1999. Subjects were censored at the date of any cancer diagnosis (except non-melanoma skin cancer), emigration, death or end of study, whichever came first. The 22 cases identified as members of the randomly drawn subcohort were analysed as cases only (weight=1), and they are included in the (weighted) Cox-likelihood at every event time they are at risk according to the Borgan II estimator.[32] Robust variance was used to compute standard errors of the HRs. The proportional hazards assumption was evaluated by Schoenfeld residuals.

Based on the literature and the available variables in NOPW, a directed acyclic graph (DAG) was drawn to help model the relationship between work schedule and breast cancer risk (online supplemental figure S1). Based on the DAG, we adjusted for age at first child, number of children, menopausal status at baseline and education in addition to attained age (timescale). Likewise, based on a DAG (online supplemental figure S2) we adjusted for age and education in the analyses of chlorinated degreasers and benzene in relation to breast cancer risk. We tested for linear trend by modelling ordinal variables as continuous variables. Tests of interaction between work schedule and chemical coexposures, and between work schedule and menopausal status were conducted by including product terms of dichotomous variables in the models.

We had up to 39% missing in the covariates of the multivariable model. We used multiple imputation with chained equations, assuming missing at random,[40] to impute 45 datasets. In each data set, we conducted analyses using the models described above, and estimates were combined using Rubin's rule.[41] We report results with and without imputed data. We have extended the description of the imputations in the supplements where we also tried an alternative imputation model (online supplemental material, methods extension, table S3).

Tests for significance were two sided, and values of p<0.05 were considered as evidence against the null hypothesis. Data analyses were performed using Stata V.16.1 (StataCorp).

## RESULTS

Table 1 shows baseline characteristics of the 86 cases and 514 non-cases in the study sample. Mean age at baseline was 45 years for cases and 42 years for non-cases. Among the cases and non-cases, mean employment duration was 8.4 and 6.9 years, the most frequent main occupational activity in last position was catering/office/administration (73% vs 71%, respectively), most had day work only (55% vs 53%) and 37% of the cases vs 42% of the non-cases were exposed to night/rollover shift. Twenty-seven per cent of the cases had >6 years of night/rollover shift compared with 18% of non-cases. Up to 68% of the cases and 77% of the non-cases may have been exposed to chlorinated degreasers. For benzene, 15% of the cases and 18% of the non-cases were exposed. Mean age at breast cancer diagnosis was 56 years, whereof 78% were ER-positive and 69% PR-positive, while 70% were HER2-negative (online supplemental table S4).

No interaction was found between work schedule and menopausal status when examining risk of overall or any breast cancer subtypes. The p values of the interaction terms in the models were between 0.279 and 0.639, we, therefore, do not present results stratified by menopausal status. Results from complete-case and multiple imputation analyses were largely similar across all analyses (tables 2–4), and we refer to the multiple imputation results in the following.

No association was found between breast cancer risk and total employment duration ($P_{trend}$=0.733) (table 2). The HR of breast cancer in relation to work schedule ever involving night/rollover shift was 0.87 (95% CI 0.52 to 1.46) vs never, and no significant association was found with duration of night/rollover shift ($P_{trend}$=0.748). Sensitivity analyses of duration of night/rollover shift by quartile among the exposed showed similar result ($P_{trend}$=0.944) (online supplemental table S5).

There was no significant association between main occupational activity in last position and breast cancer risk (table 3). No significant associations with breast cancer risk were found for chlorinated degreaser or benzene exposure (HRs (95% CIs) 0.89 (0.52 to 1.52) and 0.90 (0.46 to 1.79), respectively, for ever versus never, table 3). No significant interaction was found between work schedule and chemical co-exposures ($P_{interaction\ chlorinated\ degreasers}$=0.725; $P_{interaction\ benzene}$=0.175) (online supplemental table S6).

There was no significant association between work schedule and risk of any breast cancer subtypes. For the breast cancer subtypes that did not express sex hormone receptors we found non-significantly elevated risks for work schedule involving night/rollover shift compared with day work only (HR 1.53, 95% CI 0.50 to 4.72 for ER-negative tumours; HR 1.69, 95% CI 0.58 to 4.87 for PR-negative tumours; HR 2.22, 95% CI 0.58 to 8.51 for ER-negative and PR-negative and HER2-negative tumours) (table 4).

**Table 1** Baseline characteristics of females in the Norwegian Offshore Petroleum Workers cohort

| Variables | Cases (n=86) | Non-cases (n=514) |
|---|---|---|
| Age at baseline in 1998 (years), mean (range) | 45 (25–65) | 42 (19–74) |
| Anthropometric factors | | |
| Height (cm), mean (range)* | 167 (155–184) | 167 (151–187) |
| Weight (kg), mean (range)* | 66 (49–95) | 66 (43–163) |
| BMI (kg/m$^2$), n (%) | | |
| 12–18.4 | 1 (1) | 13 (2) |
| 18.5–24.9 | 63 (73) | 343 (67) |
| 25.0–29.9 | 15 (18) | 121 (24) |
| ≥30.0 | 5 (6) | 29 (6) |
| Missing | 2 (2) | 8 (1) |
| Reproductive history | | |
| No of children, n (%) | | |
| 0 | 29 (34) | 151 (29) |
| 1 | 24 (28) | 85 (16) |
| 2 | 13 (15) | 147 (29) |
| ≥3 | 20 (23) | 123 (24) |
| Missing | 0 (0) | 8 (2) |
| Age at first child, mean (range)* | 26 (16–38) | 24 (16–41) |
| Postmenopausal at baseline, n (%)† | 18 (21) | 111 (22) |
| Socialdemographic history | | |
| Education, n (%) | | |
| Compulsory | 18 (21) | 106 (21) |
| Vocational | 17 (20) | 118 (23) |
| Upper secondary | 23 (27) | 148 (29) |
| University/college | 24 (27) | 138 (26) |
| Missing | 4 (5) | 4 (1) |
| Work history | | |
| Main occupational activity in last position, n (%) | | |
| Production and process | 2 (2) | 23 (4) |
| Drilling and well maintenance | 3 (3) | 22 (4) |
| Maintenance/inspection/deck construction | 9 (11) | 59 (12) |
| Catering/office/administration | 63 (73) | 364 (71) |
| Miscellaneous | 9 (11) | 39 (8) |
| Missing | 0 (0) | 7 (1) |
| Total employment duration (years), mean (range) | 8.4 (0.3–20) | 6.9 (0.1–24) |
| Total employment duration quartile, n (%) | | |
| Quartile 1 (0–1.9 years) | 13 (15) | 124 (24) |
| Quartile 2 (2–5.9 years) | 23 (27) | 146 (28) |
| Quartile 3 (6–10.9 years) | 26 (30) | 121 (24) |
| Quartile 4 (11–24 years) | 24 (28) | 123 (24) |
| Work schedule, n (%) | | |
| Day work only | 47 (55) | 275 (53) |
| Mixed I (some night/ rollover+mainly day) | 1 (1) | 18 (4) |

Continued

**Table 1** Continued

| Variables | Cases (n=86) | Non-cases (n=514) |
|---|---|---|
| Mixed II (mainly night/rollover+some day) | 31 (36) | 196 (38) |
| Missing | 7 (8) | 25 (5) |
| Duration by work schedule*‡ | | |
| Day work (years), mean (range) | 6.4 (0.3–20) | 5.7 (0.1–21) |
| Night work (years), mean (range) | 3.6 (2.0–5.3) | 2.0 (0.1–11) |
| Rollover shift (years), mean (range) | 9.3 (1.0–18) | 6.8 (0.1–22) |
| Duration of night/rollover shift, n (%) | | |
| Unexposed (0 years, day work only) | 47 (55) | 275 (53) |
| ≤Median night/rollover (<1–6 years)§ | 9 (10) | 121 (24) |
| >Median night/rollover (>6 years)§ | 23 (27) | 93 (18) |
| Missing | 7 (8) | 25 (5) |
| Chemical coexposure | | |
| Chlorinated degreasers duration, n (%)¶ | | |
| Unexposed (0 years) | 28 (32) | 121 (23) |
| ≤Median (<1–5 years) | 22 (26) | 214 (42) |
| >Median (6–23 years) | 36 (42) | 179 (35) |
| Benzene duration, n (%) | | |
| Unexposed (0 years) | 73 (85) | 424 (82) |
| Exposed (<1–23 years) | 13 (15) | 90 (18) |

*Missing in continuous variables: height (n=8); weight (n=9); age at first child (n=193); day work duration (n=32); night work duration (n=32); rollover shift duration (n=32); night +rollover shift duration (n=32).
†Postmenopause assumed at ≥53 years of age.
‡Workers with complete missing information on day, night and rollover (n=32) were excluded when calculating the mean.
§Workers with day work mainly but also worked night/rollover shift were included in this category.
¶Importantly, the exposure assessment of chlorinated degreasers was probability based by experts and evaluated as 'possible' for catering workers.
BMI, body mass index.

## DISCUSSION

In this study of prospectively recruited female offshore workers, we observed no association between overall breast cancer risk and night/rollover shift, duration of such work or total employment duration. Some indication of increased risks was observed for ER-negative and PR-negative breast cancer subtypes, but these findings were not statistically significant and were based on few observations. We did not find evidence for interaction between work schedule and chemical co-exposures in relation to risk of breast cancer.

The literature is inconclusive on the association between night shift work and breast cancer risk. A report from the Nurses' Health Study[9] found an elevated risk among nurses who had long (≥20 years) cumulative night shift work and started their career early, which is also in accordance with other studies.[10 11 17] One explanation for these positive associations between long-term night shift work and breast cancer risk may be that breast tissue is particularly susceptible to circadian disruption at

a younger age.[9] However, in line with our main findings, more recent studies did not find an association between night shift work and breast cancer.[13–15] Importantly, the most recent meta-analysis, stratifying on short-term and long-term night shift work, found an increased breast cancer risk related to short-term, but not long-term night shift work.[42] Possible reasons that may explain our negative results are that, despite the extreme and unique work pattern, offshore workers have a 4-week off-duty period following each standard 2-week tour.[6] During the off-duty period, it is possible that workers could rest and readjust their circadian rhythm, and hence reduce the potential effect of night shift work on breast cancer risk. Further, the continuous shift work pattern (7–14 consecutive nights) of offshore workers might facilitate some adaptation and thereby represent a lower degree of circadian disruption, which in turn might reduce the risk of breast cancer.[5]

To our knowledge, this is the first study of night shift work and co-exposure to chlorinated degreaser and benzene in relation to breast cancer risk. The hypothesis that night shift work may potentiate chemical exposure by desynchronising detoxification mechanisms[29] was not confirmed, as we did not find any statistically significant interactions, although it may be ascribed to lack of power and misclassification of exposure.

Some studies have investigated the relationship between night shift work and breast cancer subtypes. Night shift work has been associated with sex hormone receptor-positive (ER-positive or PR-positive) tumours,[14 17 18 20] which is in line with experimental evidence that melatonin suppression enhances proliferation of ER-positive cell lines.[43] Zhu *et al*[44] suggested that melatonin disturbance could instigate two independent pathways that lead to both ER-positive and ER-negative tumours. This would be in line with our findings of increased risk of ER-negative tumours among night-shift workers, which are consistent with the study by Rabstein *et al*.[22]

Our study has several strengths. First, linkage to nationwide cancer and population registries ensured high-quality cancer data and complete follow-up. Second, we had access to independent chemical exposure estimates developed by industrial hygiene experts specifically for cancer studies in our cohort. Third, offshore workers have extreme work schedules and the NOPW cohort therefore provides particularly interesting data to study the association with breast cancer. Although, self-reported work history (ie, exposure) was collected prior to the breast cancer diagnoses, and has been shown to be robust to recall errors,[45] we cannot rule out some differential misclassification as a result of collapsing exposure categories.[46 47] However, the results of analysis with categorical and continuous exposure gave similar results (table 2). The females in the NOPW cohort were quite young at baseline (mean age 42), which may explain the relatively few cases. Also, the low case numbers in some exposure categories reduced the statistical power of detecting significant deviations from the null hypothesis.

**Table 2** HRs of female breast cancer according to work schedule in the Norwegian Offshore Petroleum Workers cohort

| Work schedule variable | Complete case analysis (n=367) | | | Multiple imputation (n=600; 86 cases) |
| --- | --- | --- | --- | --- |
| | No of participants | No of cases | HR (95% CI) | HR (95% CI) |
| Total employment duration*† | | | | |
| Quartile 1 (0–1.9 years) | 79 | 9 | 1.00 (reference) | 1.00 (reference) |
| Quartile 2 (2–5.9 years) | 99 | 16 | 1.59 (0.58 to 4.35) | 1.50 (0.69 to 3.27) |
| Quartile 3 (6–10.9 years) | 90 | 8 | 0.68 (0.24 to 1.94) | 1.64 (0.78 to 3.50) |
| Quartile 4 (11–24 years) | 99 | 16 | 1.21 (0.46 to 3.13) | 1.19 (0.54 to 2.61) |
| P-trend‡ | | | 0.864 | 0.733 |
| Work schedule involving night/rollover shift*† | | | | |
| Unexposed (day work only) | 217 | 29 | 1.00 (reference) | 1.00 (reference) |
| Exposed (night/rollover shift) | 150 | 20 | 1.06 (0.57 to 1.97) | 0.87 (0.52 to 1.46) |
| Duration of night/rollover shift*† | | | | |
| Unexposed (0 years, day work only) | 217 | 29 | 1.00 (reference) | 1.00 (reference) |
| ≤Median night/rollover (<1–6 years) | 68 | 6 | 0.72 (0.28 to 1.86) | 0.56 (0.26 to 1.22) |
| >Median night/rollover (>6 years) | 82 | 14 | 1.34 (0.67 to 2.72) | 1.21 (0.67 to 2.18) |
| P trend§ | | | 0.523 | 0.748 |

*Adjusted for age, age at first child, number of children, menopause status and education.
†Complete work history that is, up to eight employments as an offshore worker.
‡Modelled as a continuous variable to test for linear trend, HR-complete case 0.97 (95% CI 0.72 to 1.31) and HR-imputed 1.04 (95% CI 0.83 to 1.30).
§Modelled as a continuous variable to test for linear trend, HR-complete case 1.13 (95% CI 0.78 to 1.63) and HR-imputed 1.05 (95% CI 0.77 to 1.44).

Comparing our results with the pooled-analysis of Cordina-Duverger et al,[17] which had larger sample size and reported a positive association between night shift work and breast cancer (all receptor subtypes combined), the two studies do not contradict each other (HR 0.87, 95% CI 0.52 to 1.46 in our study vs OR 1.12, 95% CI 1.00 to 1.25 in Cordina-Duverger et al). Our study had relatively few cases resulting in low power and wide confidence intervals. The limited power may leave real interactions undetected. In an attempt to reduce the impact of limited power, we tested interactions with binary variables (ever/never exposure). The probability-oriented JEM for chlorinated degreasers classified catering workers as 'possibly exposed' and assigned the same exposure

**Table 3** HRs of female breast cancer according to chemical Co-Exposure in the Norwegian Offshore Petroleum Workers cohort

| Chemical exposure variable | Complete case analysis (n=367) | | | Multiple imputation (n=600; 86 cases) |
| --- | --- | --- | --- | --- |
| | No of participants | No of cases | HR (95% CI) | HR (95% CI) |
| Main occupational activity in last position* | | | | |
| Production/drilling/maintenance | 58 | 8 | 1.17 (0.47 to 2.88) | 0.92 (0.48 to 1.79) |
| Catering/office/administration | 280 | 35 | 1.00 (reference) | 1.00 (reference) |
| Miscellaneous | 29 | 6 | 1.70 (0.63 to 4.53) | 1.19 (0.54 to 2.63) |
| Chlorinated degreasers*† | | | | |
| Never | 95 | 17 | 1.00 (reference) | 1.00 (reference) |
| Ever | 272 | 32 | 0.72 (0.35 to 1.48) | 0.89 (0.52 to 1.52) |
| Benzene* | | | | |
| Never | 309 | 40 | 1.00 (reference) | 1.00 (reference) |
| Ever | 58 | 9 | 1.54 (0.65 to 3.64) | 0.90 (0.46 to 1.79) |

*Adjusted for age and education.
†Importantly, the exposure assessment of chlorinated degreasers was probability based by experts and evaluated as 'possible' for catering workers.

**Table 4** HRs of female breast cancer by receptor status subtypes in relation to work schedule in the Norwegian Offshore Petroleum Workers cohort

| Breast cancer subtype | Work schedule variable | Complete case analysis (n=367) | | | Multiple imputation (n=600; 86 cases) |
|---|---|---|---|---|---|
| | | No of participants | No of cases | HR (95% CI) | HR (95% CI) |
| ER-positive | Work schedule involving night/rollover shift* | | | | |
| | Unexposed (day work only) | 217 | 23 | 1.00 (reference) | 1.00 (reference) |
| | Exposed (night/rollover shift) | 150 | 16 | 1.00 (0.50 to 2.00) | 0.75 (0.41 to 1.36) |
| ER-negative | Work schedule involving night/rollover shift* | | | | |
| | Unexposed (day work only) | 217 | 5 | 1.00 (reference) | 1.00 (reference) |
| | Exposed (night/rollover shift) | 150 | 2 | 0.78 (0.14 to 4.46) | 1.53 (0.50 to 4.72) |
| PR-positive | Work schedule involving night/rollover shift* | | | | |
| | Unexposed (day work only) | 217 | 25 | 1.00 (reference) | 1.00 (reference) |
| | Exposed (night/rollover shift) | 150 | 13 | 0.80 (0.39 to 1.68) | 0.67 (0.36 to 1.26) |
| PR-negative | Work schedule involving night/rollover shift* | | | | |
| | Unexposed (day work only) | 217 | 3 | 1.00 (reference) | 1.00 (reference) |
| | Exposed (night/rollover shift) | 150 | 5 | 2.10 (0.55 to 8.08) | 1.69 (0.58 to 4.87) |
| HER2-positive | Work schedule involving night/rollover shift* | | | | |
| | Unexposed (day work only) | 217 | 7 | 1.00 (reference) | 1.00 (reference) |
| | Exposed (night/rollover shift) | 150 | 1 | 0.29 (0.04 to 2.14) | 0.22 (0.05 to 0.99) |
| HER2-negative | Work schedule involving night/rollover shift* | | | | |
| | Unexposed (day work only) | 217 | 21 | 1.00 (reference) | 1.00 (reference) |
| | Exposed (night/rollover shift) | 150 | 14 | 1.02 (0.49 to 2.15) | 0.88 (0.47 to 1.62) |
| ER-positive and PR-positive and HER2-negative | Work schedule involving night/rollover shift* | | | | |
| | Unexposed (day work only) | 217 | 17 | 1.00 (reference) | 1.00 (reference) |
| | Exposed (night/rollover shift) | 150 | 9 | 0.79 (0.32 to 1.95) | 0.58 (0.26 to 1.28) |
| ER-positive and PR-positive and HER2-positive | Work schedule involving night/rollover shift* | | | | |
| | Unexposed (day work only) | 217 | 5 | 1.00 (reference) | 1.00 (reference) |
| | Exposed (night/rollover shift) | 150 | 1 | 0.38 (0.05 to 2.70) | 0.29 (0.06 to 1.29) |
| ER-negative and PR-negative and HER2-positive | Work schedule involving night/rollover shift* | | | | |
| | Unexposed (day work only) | 217 | 0 | 1.00 (reference) | 1.00 (reference) |
| | Exposed (night/rollover shift) | 150 | 0 | – | – |
| ER-negative and PR-negative and HER2-negative | Work schedule involving night/rollover shift* | | | | |
| | Unexposed (day work only) | 217 | 3 | 1.00 (reference) | 1.00 (reference) |
| | Exposed (night/rollover shift) | 150 | 2 | 1.36 (0.16 to 11) | 2.22 (0.58 to 8.51) |

*Adjusted for age, age at first child, number of children, menopause status and education.
ER, oestrogen receptor; HER2, human epidermal growth factor receptor 2; PR, progesterone receptor.

level despite the fact that exposure may vary by job tasks. Hence, the actual fraction of workers exposed to chlorinated degreasers in this job-category is uncertain. We also lacked information on some potential confounders (eg, chronotype, exogenous hormone use, family history and breast feeding history), and we had no information on exposures during follow-up.

In conclusion, our findings in this prospectively recruited cohort need to be interpreted carefully due to limited statistical power, but pointed towards that night/rollover shift, exposure to chlorinated degreasers or to benzene may not be associated with increased breast cancer risk in female offshore petroleum workers. However, further studies with larger sample sizes are warranted for analyses with combined exposures and for analyses of breast cancer by receptor subtype.

**Author affiliations**
[1]Department of Research, Cancer Registry of Norway, Oslo, Norway
[2]Department of Biostatistics, University of Oslo Faculty of Medicine, Oslo, Norway
[3]Oslo Centre for Biostatistics and Epidemiology, Oslo University Hospital, Oslo, Norway
[4]Department of Epidemiology and Population Health, Albert Einstein College of Medicine, Bronx, New York, USA
[5]Department of Mathematics, University of Oslo, Oslo, Norway
[6]Department of Global Public Health and Primary Care, Universitetet i Bergen Det medisinsk-odontologiske fakultet, Bergen, Norway
[7]Occupational and Environmental Epidemiology, National Cancer Institute, Bethesda, Maryland, USA

**Contributors** JSS and TKG conceived the study. F-CL, MV, KK, TER, RG, DH, SOS, MB, JK, NR, QL, DS, MCF, RB, NS, TKG and JSS contributed to the project design. F-CL performed the data analyses. F-CL, RB and JSS performed the data management. KK contributed with their expertise on breast cancer epidemiology. MB and JK contributed to chemical exposure assessment. F-CL drafted the manuscript and all authors reviewed and revised it critically for important intellectual content and approved the final version for submission. F-CL and JSS are the guarantors.

**Funding** This work was supported by the Research Council of Norway grant number 280537.

**Competing interests** JSS and TKG have received a grant from the Research Council of Norway (governmental agency) that awarded an industry-collaborative grant to the Cancer Registry of Norway (governmental agency) in 2019 to establish a cohort of offshore petroleum workers. A condition pertaining to industry-collaborative grants is that 20% (US$175 000) of the grant was provided by the petroleum industry and 80% (US$700 000) by the Research Council itself with the intention of joining forces for the common interest of occupational health among petroleum workers. The application process was governed by the Research Council without any involvement from the industry. The grant does not cover the PIs or any of the collaborators salary.

**Patient consent for publication** Not applicable.

**Ethics approval** This study involves human participants and was approved by the Norwegian Data Inspectorate, the Regional Committee for Medical Research Ethics (2018/1162) and the Norwegian Directorate of Health.

**Provenance and peer review** Not commissioned; externally peer reviewed.

**Data availability statement** Data may be obtained from a third party and are not publicly available. The data that support the findings of this study are available from the CRN (cohort data and cancer data) and the National Population Register (death and emigration data) but restrictions apply to the availability of these data, which were used under license for the current study, and so are not publicly available. Requests for data sharing/case pooling for projects with necessary approvals and legal basis according to the EU General Data Protection Regulation (GDPR) may be directed to principal investigator Dr Tom K Grimsrud; email: tom.k.grimsrud@kreftregisteret.no

**ORCID iDs**
Fei Chih Liu http://orcid.org/0000-0002-3361-3394
Jo Steinson Stenehjem http://orcid.org/0000-0002-1964-5410

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
