## [Reviewer comments · BMJ Open]

ARTICLE DETAILS

TITLE (PROVISIONAL)	Night shift work, chemical co-exposures and risk of female breast cancer in the Norwegian Offshore Petroleum Workers (NOPW) cohort: a prospectively-recruited case-cohort study.
AUTHORS	Liu, Fei-Chih; Veierød, Marit; Kjærheim, Kristina; Robsahm, Trude; Ghiasvand, Reza; Hosgood, Dean; Samuelsen, Sven Ove; Bråtveit, Magne; Kirkeleit, Jorun; Rothman, Nat; Lan, Qing; Silverman, Debra; Friesen, Melissa C.; Babigumira, Ronnie; Shala, Nita; Grimsrud, Tom; Stenehjem, Jo

VERSION 1 – REVIEW

REVIEWER	Kartsonaki, Christiana University of Oxford
REVIEW RETURNED	19-Sep-2021

GENERAL COMMENTS	Night Shift Work, Chemical Co-exposures and Risk of Female Breast Cancer in the Norwegian Offshore Petroleum Workers (NOPW) Cohort: a prospective case-cohort study This is a case-cohort study examining the associations between night shift work, exposure to chemicals and risk of breast cancer. The data quality is high, as there is extensive information on outcomes from the Norwegian cancer registry. Although the number of breast cancer cases is relatively small, the study is interesting and worthwhile, it has been conducted in a unique population with well-defined exposures, and the results have been interpreted appropriately. There are, however, a few issues that need to be addressed: 1. 'Further, breast cancer risk in relation to night shift work may be elevated among pre-menopausal females with a positive tumour status to oestrogen receptor (ER), progesterone receptor (PR), or human epidermal growth factor receptor 2 (HER2), but the evidence is inconclusive.' This statement is unclear to me. Does it mean 'night shift work may be associated with a higher risk of ER+, PR+ or HER2+ breast cancer among pre-menopausal females?'2. In classifying breast cancer subtypes based on immunohistochemistry, I think it would be better to name the categories ER+ PR+ HER2-, ER+ PR+ HER2+, ER- PR- HER2+, and ER- PR- HER2-. I would use the luminal A/B terminology only if gene expression was used for the classification, as luminal B tumours need not be HER2+.3. Why were individuals who were diagnosed with breast cancer during follow-up excluded from the subcohort? Many case-subcohort designs rely on the subcohort being selected from the cohort at baseline, without using information that became available during follow-up. This exclusion affects the prospective nature of the study and violates the assumptions of some of the estimators for Cox regression models in case-subcohort studies. Please give the
--

	rationale for this choice and explain why this is appropriate in relation to the estimator used. 4. My understanding is that menopause status at baseline was inferred using a single age cut-off. First, was the cutoff taken directly from the Million Women Study? What is the median (and other features of the distribution) of menopause age in Norwegian women? Second, a sensitivity analysis could be done to assess to what extent this choice affects the results. 5. Which estimator from Borgan et al. was used for the analysis? Were birth cohorts taken into account in the analysis? 6. Which variables were included as explanatory variables in the imputation model? How was the study design taken into account when performing multiple imputation? 7. I would interpret p-values as strength of evidence against the null hypothesis, rather than saying 'p-values were considered to represent statistical significance'. 8. In the discussion the authors state 'we cannot rule out some differential misclassification as a result of collapsing exposure categories, possibly yielding biased HRs both towards and away from the null'. Is it not possible to not collapse exposure categories to explore this?
--	--

REVIEWER	Kang, Mo-Yeol College of Medicine, The Catholic University of Korea, Department of Occupational and Environmental Medicine
REVIEW RETURNED	22-Sep-2021

GENERAL COMMENTS	This study investigated the association between night shift work and the association of co-exposure (chlorinated degreasers and benzene) and breast cancer risk, and possible interaction with work schedule. Overall, the topic of article is important, but I think it also have critical weak point. Considering that the incidence of breast cancer is less than 100 per 100,000, the statistical power of the 600 participants in this study is very insufficient. Therefore, it is difficult to draw any conclusions from the results of this study.
---

VERSION 1 – AUTHOR RESPONSE

Reviewer: 1

Dr. Christiana Kartsonaki, University of Oxford

Comments to the Author:

Night Shift Work, Chemical Co-exposures and Risk of Female Breast Cancer in the Norwegian Offshore Petroleum Workers (NOPW) Cohort: a prospective case-cohort study

This is a case-cohort study examining the associations between night shift work, exposure to chemicals and risk of breast cancer. The data quality is high, as there is extensive information on outcomes from the Norwegian cancer registry. Although the number of breast cancer cases is relatively small, the study is interesting and worthwhile, it has been conducted in a unique population with well-defined exposures, and the results have been interpreted appropriately. There are, however, a few issues that need to be addressed:

COMMENT 1.1:

'Further, breast cancer risk in relation to night shift work may be elevated among pre-menopausal females with a positive tumour status to oestrogen receptor (ER), progesterone receptor (PR), or human epidermal growth factor receptor 2 (HER2), but the evidence is inconclusive.' This statement

is unclear to me. Does it mean 'night shift work may be associated with a higher risk of ER+, PR+ or HER2+ breast cancer among pre-menopausal females?

RESPONSE 1.1:

Thank you for pointing this out. Our intention with this sentence was to highlight that findings may differ by receptor subtype. However, we agree that this sentence was not clear and we have revised the text to clarify that findings differ by receptor subtype by stating explicitly that Rabstein et al., (ref #22) found an association with ER-negative breast cancer. The sentence now reads "Further, night shift work may be associated with an increased risk of oestrogen receptor (ER)-positive, progesterone receptor (PR)-positive or human epidermal growth factor receptor 2 (HER2)-positive breast cancer (18-21). However, night shift work has also been reported to increase the risk of ER-negative breast cancer (22), and whether the association between night shift work and breast cancer varies by subtype remains unclear." (see p5, lines3-9).

COMMENT 1.2:

In classifying breast cancer subtypes based on immunohistochemistry, I think it would be better to name the categories ER+ PR+ HER2-, ER+ PR+ HER2+, ER- PR- HER2+, and ER- PR- HER2-. I would use the luminal A/B terminology only if gene expression was used for the classification, as luminal B tumours need not be HER2+.

RESPONSE 1.2:

Thank you for the insight. We have dropped the use of luminal A, luminal B and triple negative classification in our paper, and report the combination of hormonal subtypes as your suggested. Please see the new analyses and tables (see Table 4, Table S4)

COMMENT 1.3:

Why were individuals who were diagnosed with breast cancer during follow-up excluded from the subcohort? Many case-subcohort designs rely on the subcohort being selected from the cohort at baseline, without using information that became available during follow-up. This exclusion affects the prospective nature of the study and violates the assumptions of some of the estimators for Cox regression models in case-subcohort studies. Please give the rationale for this choice and explain why this is appropriate in relation to the estimator used.

RESPONSE 1.3:

The 22 female workers who were diagnosed with breast cancer during follow-up who also were randomly drawn to subcohort, were classified as cases only with weight = 1. They were not "excluded" from the total study sample, only transferred from the subcohort to the case-group and included in the analysis using weights = 1. This is in accordance with the Borgan II estimator (ref #32: Borgan et al., 2000) which was used in the analyses. Thereby they are included in the (weighted) Cox-likelihood at every event time they are at risk, though with weights equal to one. This approach has been advocated by several authors in addition to Borgan et al (2000), such as Chen & Lo (Biometrika, 1999) and Kulich & Lin (Biometrika, 2000) and gives valid estimates. It can further be explained through post-stratification on case-status (Samuelsen, Ånestad & Skrondal, Scandinavian Journal of Statistics, 2007). Such post-stratification gives (likely only slight) efficiency gains over the other estimators discussed in Borgan et al., 2000.

COMMENT 1.4:

My understanding is that menopause status at baseline was inferred using a single age cut-off. First, was the cutoff taken directly from the Million Women Study? What is the median (and other features of the distribution) of menopause age in Norwegian women? Second, a sensitivity analysis could be done to assess to what extent this choice affects the results.

RESPONSE 1.4:

Yes, the cut-off of age ≥ 53 was based on the convention from the Million Women Study (MWS), and later adapted in the Norwegian Women and Cancer (NOWAC) study for women with missing data on menopause (ref #38: Busund et al., 2018). NOWAC is a large population-based prospective study established in 1991 representative of the Norwegian female population, including 165,772 women aged 30–70 years at recruitment (Lund et al. *Int J Epidemiol* 2008;37:36-41, <https://doi.org/10.1093/ije/dym137>). In addition, a cross-sectional study within the NOWAC postgenome cohort found that 90% of postmenopausal women experiencing natural menopause were postmenopausal by the age of ≥ 53 (Waaseth et al. *BMC Women's Health* 2008; 8:1, doi:10.1186/1472-6874-8-1). Furthermore, a study of temporal trends in age at menopause based on the Norwegian breast cancer screening program (BreastScreen Norway) reported that the mean age at natural menopause increased from 50.31 years among women born during 1936-1939 to 52.73 years among women born during 1960-1964 (Gottschalk et al. *Hum Reprod* 2020;35(2):464-71, doi: 10.1093/humrep/dez288).

On the other hand, another study based on the BreastScreen Norway reported the median age at menopause was ≥ 51 years (ref#39: Bjelland et al., 2018). We therefore performed sensitivity analyses using 51 years as age cut-off for menopause status, and presented result for hazard ratios of female breast cancer according to work schedule (see table S1) and by receptor status subtypes in relation to work schedule (see table S2).

The results of the analyses using ≥ 51 as cut-off did not change materially compared to those using age ≥ 53 as cut-off, and we therefore chose to report main results with age ≥ 53 as cut-off point. Also, with 71% of the female offshore workers born after 1955 (ref #30: Stenehjem et al., 2020) a cut-off at age ≥ 53 was closer to the increase in postmenopausal age among the younger birth cohorts reported by Gottschalk et al. (*Hum Reprod* 2020;35(2):464-71, doi: 10.1093/humrep/dez288).

We now clarified in the manuscript that this choice of cut-off is also applied in the Norwegian female population (see p10, lines22-25; p11, lines1-2).

COMMENT 1.5:

Which estimator from Borgan et al. was used for the analysis? Were birth cohorts taken into account in the analysis?

RESPONSE 1.5:

As discussed above the Borgan II estimator was used for analysis where we assigned a weight to all subcohort non-cases based on the inverse sampling fraction of their corresponding 5-year birth cohort stratum to ensure that comparable non-cases were available for all cases. Specifically, we selected all eligible female cohort members that were born in 5-year birth cohorts 1920-1924, 1925-1929, 1930-1934, 1935-1939, and 1975-1979 in order to secure that the oldest and the youngest female non-cases were comparable to the cases in these birth cohorts. Then, for those born 1940-1974, we drew at random within the 5-year birth cohorts and calculated a sampling fraction, defined as random sample / total cohort member, for each birth cohort. We then assigned a weight, which was defined as $1/\text{sampling fraction}$, for each birth cohort and those weights were specified as offset in in the Cox regression models. Robust variance was used to obtain standard errors. We have also tried model-based variances previously, but found little difference compared to those obtained by robust variance.

COMMENT 1.6:

Which variables were included as explanatory variables in the imputation model? How was the study design taken into account when performing multiple imputation?

RESPONSE 1.6:

The explanatory variables that entered the imputation models were the same as those that entered the complete cases models. The Cox PH model was specified exactly the same when ran on the imputed dataset as when ran on the complete case dataset, with weights as offset and with standard errors derived from robust variance. We also wish to add that to arrive at the imputed dataset we combined predicting variables (i.e. date of birth, total employment duration, and breast cancer diagnosis) with variables from the estimation model (i.e. work schedule involving night/rollover shift, duration of night/rollover shift, number of children, age at first child, education, and main occupational activity in last position) to simultaneously predict missing data on all variables according to the multiple imputation with chained equations approach as described by White et al., 2011 (White IR, Royston P, Wood AM. Multiple imputation using chained equations: Issues and guidance for practice. *Statistics in medicine* 2011; 30: 377-99.). This has been described in the methods' extension in the supplemental material, it reads "We identified differences between these samples for date of birth, total employment duration, and breast cancer diagnosis, and used these variables as predictors together with the variables from the estimation model (i.e. work schedule involving night/rollover shift, duration of night/rollover shift, number of children, age at first child, education, and main occupational activity in last position) to simultaneously predict missing data on all variables. Linear and multinomial logit models were used to predict missing data for continuous and categorical variables, respectively." (see Supplemental material p2, lines22-25; p3, lines1-3).

COMMENT 1.7:

I would interpret p-values as strength of evidence against the null hypothesis, rather than saying 'p-values were considered to represent statistical significance'.

RESPONSE 1.7:

Thank you for the valuable suggestion, we have amended the text accordingly (see p12, lines15-16).

COMMENT 1.8:

In the discussion the authors state 'we cannot rule out some differential misclassification as a result of collapsing exposure categories, possibly yielding biased HRs both towards and away from the null'. Is it not possible to not collapse exposure categories to explore this?

RESPONSE 1.8:

The harmonised work schedule raw data (duration of day, night and rollover) were continuous variables, and it is possible to keep them as continuous variables as we did in the trend-analysis for total employment duration and duration of night/rollover shift (Table 2, footnote C; Table S4, footnote D). We chose to collapse the exposure categories into never/ever, medians and quartiles for two main reasons: 1) to examine possible variation in risk for different magnitudes of exposure, and 2) to allow for comparison with other studies, which have reported relative risks of breast cancer according to never/ever exposure to night shift work. Importantly, both the continuous and the categorical analysis yielded similar results, and we have updated the tables with the HRs and 95% CIs from the trend analyses (See Table 2, footnotes c1, c2; Table S3, footnotes d1, d2). We have shortened the sentence about misclassification due to categorization, and included a sentence about the similar results with the two approaches (see p16, lines6-7).

Reviewer: 2

Prof. Mo-Yeol Kang, College of Medicine, The Catholic University of Korea

Comments to the Author:

This study investigated the association between night shift work and the association of co-exposure (chlorinated degreasers and benzene) and breast cancer risk, and possible interaction with work schedule. Overall, the topic of article is important, but I think it also have critical weak point.

Considering that the incidence of breast cancer is less than 100 per 100,000, the statistical power of the 600 participants in this study is very insufficient. Therefore, it is difficult to draw any conclusions from the results of this study.

RESPONSE TO REVIEWER#2:

Thank you for your important comment. We agree and did acknowledge in the Discussion that our study has limited statistical power (see page16, lines8-17). We have now modified the conclusion at the end of the discussion to make it more clear that our findings need careful interpretation due to the limited number of cases (see page16, line25; page17, lines1-3). Still, as pointed out by Reviewer 1, we respectfully think that the following points make our study worthwhile to publish:

- 1) The topic is of current interest and limited evidence exists concerning the potential influence of co-exposures at work, when examining the association between night shift work and breast cancer.
- 2) The cohort is unique in terms of exposure and despite the small size (n=2570), the follow-up is long (18.5 years).
- 3) We have transparently and carefully handled and analysed the data with appropriate and well-described methods, and sought to honestly and prudently interpret the results and discuss the limitations to not deceive or mislead the reader.
- 4) Despite the limited statistical power, the results rule out large effects in this cohort with extreme exposure.
- 5) We wish to clarify that all 86 cases in the cohort of 2570 female workers were included in the analyses. Therefore, the statistical power of this case-cohort study (86 cases and 514 subcohort non-cases) is virtually equal to that of the main cohort of 2570 female workers. Also, the case (n=86) non-case (n=514) ratio is above 1:5, which secures adequate statistical efficiency for the case-cohort dataset.

VERSION 2 – REVIEW

REVIEWER	Kartsonaki, Christiana University of Oxford
REVIEW RETURNED	24-Nov-2021

GENERAL COMMENTS	The authors have adequately addressed previous comments. I suggest to include in the statistical methods section (perhaps in the supplementary material) the details provided in response to some of my questions (e.g. estimator used and how cases arising in the subcohort were handled in the analysis), which were very thorough, such that if one were to repeat the analysis they would be able to do so based on the information provided in the paper.
---

VERSION 2 – AUTHOR RESPONSE

Reviewer: 1

Dr. Christiana Kartsonaki, University of Oxford

COMMENTS:

The authors have adequately addressed previous comments. I suggest to include in the statistical methods section (perhaps in the supplementary material) the details provided in response to some of my questions (e.g. estimator used and how cases arising in the subcohort were handled in the analysis), which were very thorough, such that if one were to repeat the analysis they would be able to do so based on the information provided in the paper.

RESPONSE:

Thank you for providing insightful comments in the previous round that helped us to improve the manuscript, we are happy that you found them to be adequately addressed. We agree that including what estimator was used and how cases in the subcohort were handled, will improve the reproducibility of this paper's results and be helpful. We have amended the manuscript accordingly. Please see p8, lines2-3; p11, lines 6-10; p11, lines14-15.